# The Impact of Mineral and Energy Supplementation and Phytogenic Compounds on Rumen Microbial Diversity and Nitrogen Utilization in Grazing Beef Cattle

**DOI:** 10.3390/microorganisms11030810

**Published:** 2023-03-22

**Authors:** Ronyatta Weich Teobaldo, Yury Tatiana Granja-Salcedo, Abmael da Silva Cardoso, Milena Tavares Lima Constancio, Thais Ribeiro Brito, Eliéder Prates Romanzini, Ricardo Andrade Reis

**Affiliations:** 1Department of Animal Science, São Paulo State University “Júlio de Mesquita Filho” (UNESP), Jaboticabal 14887-900, Brazil; 2Corporación Colombiana de Investigación Agropecuaria (AGROSAVIA), Centro de Investigación El Nus, San Roque 053030, Colombia; 3Ona Range Cattle Research and Education Center, University of Florida, Ona, FL 33865, USA; 4Department of Technology, São Paulo State University “Júlio de Mesquita Filho” (UNESP), Jaboticabal 14887-900, Brazil

**Keywords:** digestibility, essential oils, hydrolyzable tannins, rumen fermentation, ruminal bacteria

## Abstract

The objective of this study was to evaluate the effect of the addition of a phytogenic compound blend (PHA) containing hydrolyzable tannins, carvacrol, and cinnamaldehyde oil to mineral salt or energy supplementation on the rumen microbiota and nitrogen metabolism of grazing Nellore cattle. Eight castrated Nellore steers were distributed in a double-Latin-square 4 × 4 design, with a 2 × 2 factorial arrangement (two types of supplements with or without the addition of the PHA), as follows: energy supplement without the PHA addition (EW); energy supplement with the PHA addition (EPHA); mineral supplement without the addition of the PHA (MW); mineral supplement with the PHA addition (MPHA). Steers that received supplements with the PHA have a lower ruminal proportion of valerate (with the PHA, 1.06%; without the PHA, 1.15%), a lower ruminal abundance of Verrucomicrobia, and a tendency for lower DM digestibility (with the PHA, 62.8%; without the PHA, 64.8%). Energy supplements allowed for higher ammonia concentrations (+2.28 mg of NH_3_-N/dL), increased the propionate proportion (+0.29% of total VFA), and had a higher ruminal abundance of Proteobacteria and Spirochaetae phyla in the rumen. The PHA addition in the supplement did not improve nitrogen retention, reduced the ruminal proportion of valerate, and had a negative impact on both the total dry-matter digestibility and the abundance of several ruminal bacterial groups belonging to the Firmicutes and Verrucomicrobia phyla.

## 1. Introduction

The activity of the ruminal microbiota is crucial for supplying nutrients to the host, and there is a significant association between diet utilization efficiency and rumen microbial diversity [1]. While antimicrobial feed additives have the potential to improve feed efficiency by modulating rumen fermentation, their use in ruminant nutrition has been banned due to concerns about the possible generation of bacterial resistance and residues in animal products [2,3]. Therefore, phytogenic compounds such as tannins and essential oils emerge as safer alternatives for modulating the ruminal environment and improving the efficiency of diet utilization in cattle.

Hydrolyzable tannins are water-soluble polyphenolic compounds that do not bind proteins in the rumen. However, they can be broken down into monomeric subunits of low molecular weight with antimicrobial potential when they react with secreted extracellular enzymes and microbial cell walls [4]. Tannin extracts can affect the rumen microbiota by reducing the diversity and increasing the richness of the ruminal bacteria population [5], potentially altering the rumen fermentation of the diet [6,7]. Thus, low-to-moderate inclusion levels (≤20 g/kg DM) of hydrolyzable tannins have been used in cattle diets [8,9,10] to avoid the toxicity caused by the degradation of tannins by some rumen microorganisms [11] and to obtain beneficial effects, such as a higher concentration of ruminal volatile fatty acids [12], a lower acetate: propionate ratio [13], and reduced nitrogen losses in urine [14].

Compounds found in essential oils, such as cinnamaldehyde derived from cinnamon, have shown high antioxidant power and antimicrobial activity [15] due to the presence of chemical compounds belonging to the phenylpropanoids class. These compounds can damage the bacterial enzymes responsible for energy production and denature proteins [16]. Cinnamaldehyde may enhance energy efficiency and nitrogen utilization in the rumen [17], thereby reducing the *Prevotella* spp. population [18]. Carvacrol is a phenolic compound from oregano with nonspecific antimicrobial activity because of the hydroxyl group present in its molecule, which functions as an ion transporter, and its high hydrophobicity [18]. Carvacrol can influence Gram-positive and Gram-negative bacteria and modulate ruminal fermentation, reducing ruminal protein degradability and the acetate: butyrate ratio [18,19]. The association of essential oils has demonstrated promising effects on ruminal fermentation modulation based on the dosage and basal diet [20]. A blend of essential oils, bioflavonoids, and tannins has been found to enhance the diet digestibility and the feed conversion rate in dairy cattle [21]. However, the combination of hydrolyzable tannins and essential oils during the supplementation of grazing beef cattle had a negative impact on forage intake and digestibility [22].

In Brazil, beef cattle production relies on tropical grasses and mineral supplementation. However, during the rainy season, forages reach their peak production and nutritional quality, requiring protein: energy ratio balancing through supplementation. Supplementing energy in well-managed tropical pasture-based systems can improve the uptake of nitrogen compounds in the rumen, enhance ruminal fermentation, and increase digestibility in grazing cattle [23]. Additionally, adding a phytogenic compound blend to the supplement could further enhance the efficiency of nitrogen utilization [14,21]. However, the response of the rumen microbiota and nitrogen metabolism in steers grazing tropical forages when supplemented with energy and phytogenic compounds has not yet been reported. Therefore, it was hypothesized that the use of a blend of hydrolyzable tannins and essential oils in the energy supplement of grazing steers could improve nitrogen metabolism by modulating the ruminal microbiome and reducing nitrogen excretion via the urine. The objective of this study was to evaluate the effect of including a phytogenic compound blend (PHA) containing hydrolyzable tannins, carvacrol, and cinnamaldehyde oil supplied at a dose of 1.5 g/kg of ingested dry matter (DM) in the mineral salt or energy supplementation on the rumen microbiota and nitrogen metabolism of grazing Nellore cattle during the rainy season.

## 2. Materials and Methods

### 2.1. Grazing Area and Animals

The experiment was carried out at the Faculty of Agricultural and Veterinary Sciences (FCAV) of the São Paulo State University “Júlio de Mesquita Filho” (UNESP), Jaboticabal campus, São Paulo (Brazil), during the rainy season from January to April 2018 (21°15′22″ S latitude and 48°18′58″ W longitude). The climate of this region is the subtropical type AW according to the Köppen classification. Animal care and handling followed the guidelines of the Brazilian College of Animal Experimentation (COBEA) and was approved by the Ethics Committee on the Use of Animals (CEUA) of FCAV/UNESP (protocol No. 12703/15).

Eight castrated Nellore steers with an average body weight (BW) of 456.6 ± 32.8 kg and fitted with silicone-type cannula in the rumen were randomly distributed in a double-Latin-square 4 × 4 design, with a 2 × 2 factorial arrangement of treatments (energy or mineral supplement with or without phytogenic compounds) and four experimental periods. Steers were housed in eight paddocks of 0.7 and 1.3 ha (one steer in each paddock) formed by *Urochloa brizantha* cv. Marandu, delimited by electric fencing, and served by a trough with access from both sides (30 cm linear per animal) and a drinking fountain. The meteorological data and maintenance of fertilization in the experimental area during the experimental period were described by Teobaldo et al. [22]. Pastures were managed under continuous stocking with variable stocking rates and a canopy target of approximately 25 cm high, using the “put-and-take” technique [24], and animals’ regulators to adjust and maintain the height target, in addition to the steer testers. The forage height was measured weekly from 80 random points per paddock to the stocking rate adjustment. Every 28 days, three points per paddock were sampled (using a frame of 0.25 m^2^), representing the average forage height, to evaluate the forage mass and morphological components; thus, the average total herbage mass was 6.52 t of DM per ha with an average herbage allowance of 3.86 kg of DM per kg of BW.

The steers were distributed into the paddocks two weeks before the beginning of the experiment to adapt to the conditions. Then, four experimental periods of 28 days were conducted, considering 14 days of treatment adaptation and 14 days of the sampling phase, to test four supplements: energy supplement without the phytogenic compound addition (EW); energy supplement with the phytogenic compounds addition (EPHA); mineral supplement without the addition of phytogenic compounds (MW); mineral supplement with the addition of phytogenic compounds (MPHA) (Table 1). Both energy supplements (EW and EPHA) were offered daily at 09:00 a.m. at 0.3% BW, while mineral supplements were provided ad libitum. The phytogenic compound blend (PHA) contained 10% of carvacrol and cinnamaldehyde oil, and 90% hydrolyzable tannins extracted from berries and grapes, supplied at a dose of 1.5 g/kg of ingested DM, as recommended by commercial suppliers. The energy supplements used were commercial, did not contain urea, and contained minerals (Ca, 20.0 g/kg; Na, 20.0 g/kg; S, 10.0 g/kg; Cu, 133.0 mg/kg; Mn, 49 mg/kg; Zn, 340.0 mg/kg; Co, 1.4 mg/kg; I, 7.0 mg/kg; Se, 2.25 mg/kg). The suppliers were not authorized to disclose the percentage composition of the ingredients used. Mineral salt supplements had guaranteed levels: Ca, 123.0 g/kg; P, 90.0 g/kg; Cu, 1040.0 mg/kg; Mn, 500 mg/kg; Zn, 2000.0 mg/kg; Co, 15.0 mg/kg; I, 67.0 mg/kg; Se, 14.0 mg/kg.

### 2.2. Intake and Digestibility

The intake and digestibility of the forage were evaluated on days 16–28 of each experimental period using Chromium oxide (Cr_2_O_3_) as an external marker to estimate the excretion of fecal matter and the iNDF as an internal marker. One cellulose capsule containing 10 g of chromium oxide was placed in the rumen of each animal through the cannula, daily at 9:00 for 10 days (7 days of adaptation and 3 days of fecal sampling). Fecal samples were collected directly from the rectum once daily, alternating at the following times: 5:00 p.m., 11:00 a.m., and 6:00 a.m. The forage bromatological composition was evaluated in forage samples collected by the manual grazing simulation method [25] each 28 d. The average of each supplement intake was calculated from the average amount of supplement provided (% BW) per paddock.

Forage, energy supplements, and feces samples were dried under forced air at 55 °C for 72 h and ground in a Willey mill with a 1 or 2 mm mesh sieve for bromatological and indigestible neutral detergent fiber analysis (iNDF), respectively. Then, the DM, organic matter (OM), mineral matter (MM), and ether extract (EE) were determined according to AOAC [26]. The crude protein (CP) was determined using the Dumas method in a LECO FP-528 N analyzer (Leco Corporation, St. Joseph, MI, USA). Neutral (NDF) and acid (ADF) detergent fiber were determined following the recommendations of Van Soest et al. [27] and Goering and Van Soest [28], respectively, using the Ankom^®^ 2000 equipment (Ankom Technologies, Fairport, NY, USA). The α-amylase was added during the NDF procedures of the energy supplements samples. In situ incubation procedure for 288 h [29] was used to obtain the iNDF content. The gross energy (GE) content was obtained by the combustion of samples in an adiabatic calorimetric pump (PARR Instrument Company 6300, Moline, IL, USA). Chromium oxide in feces was measured after acid digestion using an atomic-absorption spectrophotometer [30].

### 2.3. Nitrogen Metabolism

Urine spot samples were collected once daily at the same moment as feces collection, during steer spontaneous urination. Approximately 50 mL of urine were filtered through three layers of cheesecloth, and two aliquots were immediately stored at −20 °C for later analysis of creatinine, uric acid, and urea by a colorimetric–enzymatic method using commercial kits (Labtest^®^, Lagoa Santa, MG, Brazil), the total N concentration through the Dumas method in a LECO FP-528 N analyzer (Leco Corporation, St. Joseph, MI, USA), and allantoin by the colorimetric method as described by Chen and Gomes [31].

Daily urinary volume was estimated according to Costa e Silva et al. [32] based on the relationship between daily urinary creatinine excretion and BW. Then, the daily urinary excretion of N compounds (urea and total nitrogen) was calculated as the product of the total N compound concentration in the samples and the estimated daily urinary volume. The daily N intake was obtained as a summatory of the N intake from forage and the N intake from the supplement. The daily fecal excretion of N was calculated as the de product of the total N concentration in feces and the estimated excretion of fecal matter.

The total purine derivates were calculated as the summation of allantoin and uric acid and expressed in mmol/d [30]. Purine derivates absorbed (Y, mmol/d) were calculated using the equation proposed by Verbic et al. [33]: Y = (X − (0.30 × BW0.75))/0.80, where 0.80 corresponds to the recovery of absorbed purines as purine derivatives in the urine (mmol/mmol) and 0.30 × BW0.75 is the endogenous excretion of purine derivatives (mmol) in the urine per unit of metabolic body size [33]. The ruminal microbial nitrogen synthesis (Nmic, g N/day) was estimated using the equation proposed by Barbosa et al. [34]: Y = 70 × X/(0.83 × 0.137 × 1000), where 70 is the N content of purines (mg/mmol), 0.137 corresponds to the purine N: total N ratio in bacteria, and 0.83 is the true digestibility of microbial purines. The digestible organic matter apparently fermented in the rumen (OMFR) was calculated as the product of the digestible OM intake and the factor 0.65 [35], and later used to calculate the efficiency of microbial protein synthesis, expressed in g of Nmic/kg of OMFR.

Blood samples were collected before supplementation (0 h) and 4 h after supplementation on day 28 of each experimental period. Blood was sampled directly from the caudal vein using vacuum tubes and clotting accelerator gel, and stored at −20 °C to determine the serum urea level later through commercial kits (Labtest^®^, Lagoa Santa, MG, Brazil). The conversion of urea values into urea N was obtained as the product of values obtained by commercial kits and the factor 0.466.

### 2.4. Rumen Fermentation Parameters and Microbiota Population

Rumen content was collected through the ruminal cannula on day 20 of each experimental period at 0, 6, 9, and 12 h after supplementation and immediately filtered through double layers of cheesecloth; then, approximately 100 mL of ruminal fluid was recovered. The pH was measured by a digital pH meter (TEC 7, Tecnal, Piracicaba, SP, Brazil) and two aliquots of approximately 50 mL were stored at −20 °C. Later, one aliquot was used for ruminal ammonia (NH_3_-N) concentration analysis in a Kjeldahl system [26], and the second aliquot was used for volatile fatty acid (VFA) quantification following the recommendations of Famme and Knudsen [36] by gas chromatography (2014AF, Shimatzu Corporation, Kyoto, Japan).

Approximately 100 g of rumen content (solid + liquid) was collected from several regions of the rumen before supplementation (0 h) on day 20. These samples were immediately stored in coolers with ice and transported to the laboratory. Then, a bacterial pellet was obtained as described by Granja-Salcedo et al. [37]. Briefly, 60 g of the rumen contents were mixed with 60 mL of phosphate saline buffer (pH 7.4) and stirred for 3 min. The mixture was then filtered with a mesh fabric (100 μm) to remove any solid particles. The resulting filtrate was then subjected to centrifugation at 16,000× *g* for 10 min at 4 °C. The supernatant was discarded and the remaining pellet was resuspended in 4 mL of tris-EDTA buffer (10×, pH 8.0). The resuspended content was then subjected to centrifugation again at 16,000× *g* for 10 min at 4 °C, and the supernatant was again discarded. The DNA extraction was carried out using the Quick-DNA™ Fecal/Soil Microbe Miniprep kit (Zymo Research Corporation, CA, USA) and the bead-beating method for cell lysis (FastPrep-24, MP, Biomedicals, Illkirch, France). The DNA concentrations were checked by a spectrophotometer (NanoDropR ND-1000 Spectrophotometer, Thermo Fisher Scientific, Waltham, MA, USA) and fluorometer (QubitR3.0, kit Qubit RdsDNA Broad Range Assay Kit, Life Technologies, Carlsbad, CA, USA). The DNA purity was checked through absorbance ratios (260/230 and 260/280 nm), and DNA integrity was assessed by 0.8% (*w/v*) agarose gel electrophoresis stained with SYBR Safe DNA Gel Stains (Thermo Fisher Scientific, Waltham, MA, EUA).

Libraries in duplicate were prepared by PCR amplification of the V4–V5 regions of the 16S ribosomal RNA gene 16S rRNA using barcoded 16S Illumina primers (515F and 926R) [38]. PCR product length and amplicon size were checked by 1% (*w/v*) agarose gel electrophoresis using a 1 kb plus DNA ladder (Invitrogen, Carlsbad, CA, USA). Then, PCR fragments were purified using a Zymoclean TM Gel DNA Recovery kit (Zymo Research Corporation, CA, USA). Sequencing was performed using the MiSeq Reagent v2 (2 × 250 bp; Illumina^®^, USA) kit in an Illumina MiSeq^®^ Machine.

Bioinformatics analyses were performed with QIIME 2 [39]. Raw sequence data were demultiplexed and quality-filtered using the q2-demux plugin followed by denoising with DADA2 [40]. The q2-diversity plugin was used to estimate the diversity metrics (after the samples were rarefied (subsampled without replacement) to 4994 sequences per sample): alpha-diversity metrics (observed OTUs, evenness, and Faith’s Phylogenetic Diversity [41] and beta-diversity metrics (unweighted UniFrac [42] and Bray–Curtis dissimilarity). Taxonomy was assigned to ASVs using the q2-feature-classifier classify-consensus-vsearch taxonomy classifier against the Silva 128 database with 97% OTU reference sequences [43].

### 2.5. Statistical Analyses

Intake, digestibility, and nitrogen metabolism data were analyzed using the PROC MIXED in SAS v.9.2 (SAS Inst. Inc., Cary, NC, USA) in a double 4 × 4 Latin-square design with a factorial arrangement (A × B), where the fixed effect of factor A corresponded to the type of supplement as mineral or energy supplement, and the factor B to the phytogenic compound blend (PHA) addition (yes or no). The ANOVA also included the interactions of factors (A × B), treatments error, the random effects of the Latin square, period, animal, the period × animal interaction, and residues corresponding to the model. Tukey’s post hoc test was applied when the ANOVA indicated a significant difference.

Data on rumen fermentation parameters were analyzed as repeated-measures ANOVA in a double 4 × 4 Latin-square design with a split-plot factorial arrangement (A × B). The model included the fixed effects of factor A and factor B that were considered as independent variables; the sampling times considered as the dependent variable (covariate), interactions of factors (A × B), interactions of factors and time, and the random effects of the Latin square, period, animal, the period × animal interaction, and residues corresponding to the model, and the Tukey’s post hoc test was applied when the ANOVA indicated a significant difference.

Microbiota population data were compared between factor A and factor B groups by the Friedman rank test. Interactions of factors were evaluated using the Kruskal–Wallis and Dunn’s post hoc tests using R software version 4.1.1 (R Core Team, 2015).

For all tests, a probability of *p* ≤ 0.05 was considered significant and tendencies were considered when 0.05 < *p* ≤ 0.10.

## 3. Results

### 3.1. Intake and Digestibility

No effects of the interaction between the type of supplement and addition of the PHA were observed on the intake and apparent total digestibility (Table 2). However, steers supplemented with energy supplements showed a significant increase in total DM intake, expressed as the % of BW or as kg/d, as well as in the intake of OM, CP, neutral detergent fiber corrected for ash and protein (apNDF), gross energy (GE), digestible energy, and metabolizable energy compared to steers supplemented with mineral supplements (*p* < 0.05). Additionally, steers supplemented with energy supplements showed a tendency towards increased forage intake, expressed as the % of BW (*p* = 0.059) or kg/d (*p* = 0.070), compared to those supplemented with mineral supplements.

Steers supplemented with energy supplements showed a higher apparent total digestibility DM, OM, CP, apNDF, and GE than those supplemented mineral supplements (*p* ≤ 0.01). Furthermore, a tendency for the lower total digestibility of DM was observed when the PHA was added to the supplements, with steers fed supplements without the PHA exhibiting a higher digestibility (64.8%) than those fed supplements with the PHA (62.8%) (*p* = 0.073).

### 3.2. Nitrogen Metabolism

There was a type of supplement × addition of the PHA interaction effect on the amount of urinary nitrogen excreted (Table 3; *p* = 0.033). Thus, steers supplemented with the EW showed higher values of urinary nitrogen excretion (g of N per day) compared to those supplemented with the MW and MPHA supplements.

Steers supplemented with energy supplements showed a higher N intake, higher N retained, expressed as the g/d and as the % of the N intake, and a higher plasma urea N compared to steers supplemented with mineral supplements (*p* < 0.05). The plasma urea N concentration increased from 11.1 mg/dL at 0 h to 16.43 mg/dL at 4 h after supplementation (*p* < 0.001). The fecal N excreted tended to increase in steers supplemented with energy supplements (*p* = 0.081). The efficiency of microbial protein synthesis expressed as g Nmic/kg OMFR and as g Pmic/kg DOM tended to increase in steers supplemented with mineral supplements compared to those supplemented with energy supplements (*p* = 0.055).

### 3.3. Rumen Fermentation Parameters

No significant effects or tendencies were observed for the type of supplement × the addition of the PHA, and type of supplement × the addition of the PHA × time interactions on ruminal fermentation parameters (Table 4; *p* > 0.10). Energy supplements increased the concentrations of ruminal ammonia (16.51 vs. 14.23 mg of NH_3_-N/dL; *p* = 0.003) and the proportion of ruminal propionate (16.89 vs. 16.60%; *p* = 0.010). Steers that received supplements with the PHA showed a lower ruminal proportion of valerate (with the PHA, 1.06% vs. without the PHA, 1.15%; *p* = 0.015) and tended to have a higher ruminal proportion of propionate (with the PHA, 16.80% vs. without the PHA, 16.69%; *p* = 0.074) than steers fed supplements without the PHA. There was an interaction effect of the type of supplement × time on the ruminal proportion of acetate, butyrate, valerate, and isovalerate, as well as on the acetate: propionate ratio (Table 4; *p* < 0.05). Thus, at 3 h after supplementation, steers fed energy supplements showed a lower proportion of acetate and a lower acetate: propionate ratio, as well as higher proportions of valerate, isovalerate, and butyrate in the rumen than steers supplemented with mineral supplements (Figure 1).

### 3.4. Ruminal Microbiota Population

The richness index (ACE and Chao 1) and diversity estimators (Fisher, Simpson, and Shannon) of the ruminal microbial population were not affected by the supplements (*p* > 0.10; Appendix A). However, the beta-diversity Bray–Curtis distance analysis revealed that rumen microbial communities differed based on the type supplement used (*p* = 0.015; Figure 2).

Eleven phyla were identified (Table 5), and 12.66 ± 2.05% of the OTUs identified in each sample could not be classified at the phylum level. Only the phylum Euryarchaeota was identified in the Archaea domain, and it was not influenced by the supplements tested. Steers supplemented with energy supplements had higher ruminal abundance of Proteobacteria and Spirochaetae phyla compared to those supplemented with mineral supplements (*p* < 0.05). Additionally, steers supplemented with energy supplements tended to have a higher ruminal abundance of Bacteroidetes (*p* = 0.089) and a lower ruminal abundance of Firmicutes (*p* = 0.069) than those supplemented with mineral supplements. Furthermore, steers supplemented with the PHA exhibited lower ruminal abundance of Verrucomicrobia phylum than those fed supplements without the PHA (*p* = 0.022).

At the family level, 16 bacterial families were identified, with the most abundant being Lachnospiraceae (17.73%), Ruminococcaceae (14.90%), and Prevotellaceae (13.46%). The relative abundance of the Bacteroidales S24-7 group and the Veillonellaceae families in the rumen was influenced by the interaction between the type of supplement provided and the use of the PHA (Figure 3A). Specifically, the relative abundance of the Veillonellaceae family was lower in steers supplemented with the EPHA compared to those supplemented with other supplements (*p* = 0.007), and the Bacteroidales S24-7 group (*p* = 0.084) tended to be lower in steers supplemented with the MW. In addition, steers supplemented with the PHA showed a higher ruminal abundance of the Bacteroidales BS11 gut group (*p* = 0.024).

The abundance of three Archaea OTUs in the rumen was influenced at the genera level by the interaction type of supplement × the addition of the PHA (Figure 3B). The relative abundance of *Methanosphaera* was lower in the rumen of steers supplemented with the MW when compared to other supplements (*p* = 0.041). The abundance of Thermoplasmatales Incertae Sedis was lower in the rumen of steers supplemented with the MW compared to the EW and EPHA supplements (*p* = 0.011). Although an *uncultured archaeon* from Thermoplasmatales was not identified in the rumen of steers supplemented with the MPHA, its abundance was lower in the rumen of steers supplemented with the MW compared to the EW and EPHA supplements (*p* = 0.018).

The interaction between the type of supplement × the addition of the PHA influenced several bacterial OTUs at the genera level (Figure 3C). The abundance of both *Roseburia* (*p* = 0.033) and *Family XIII UCG-002* (*p* = 0.034) was lower in the rumen of steers supplemented with the MPHA when compared to those supplemented with the EW and MW, respectively. The relative abundance of the genus *Moryella* was higher in steers supplemented with the EW than in the other groups (*p* = 0.044). The ruminal abundance of *Succinivibrionaceae UCG-002* was lower in steers supplemented with MW when compared to both the energy supplements EW and EPHA (*p* = 0.045). A higher ruminal abundance of *SR1 Ambiguous taxa* was observed in the rumen of steers supplemented with both mineral supplements (MW and MPHA) when compared to those supplemented with the EPHA (*p* = 0.017). The relative abundance of the OTU class *WCHB1-41 uncultured* rumen bacterium was higher in the rumen of steers supplemented with the EW than in those supplemented with both PHA supplements (*p* = 0.041). The relative abundance of *Prevotellaceae UCG-003* tended to be lower in the rumen of animals that received the MPHA than in those that received the MW and EPHA (*p* = 0.071). *Bacteroidales RF16 group uncultured* (*p* = 0.060) and *Ruminococcaceae uncultured* (*p* = 0.071) tended to be lower in the rumen of animals supplemented with the EW than in those that received the EPHA and MW. A tendency of a lower ruminal abundance of the genus *Prevotellaceae NK3B31 group* (*p* = 0.076) and *Treponema 2* (*p* = 0.058) was also observed in the MW than in those supplemented with the EW.

In addition, energy supplements allowed a lower ruminal abundance of *Butyrivibrio 2* (*p* = 0.033), *Mollicutes RF9 uncultured rumen bacterium* (*p* = 0.038), and *Ruminococcaceae UCG-014* (*p* = 0.040), and an increase in the abundance of *Papillibacter* (*p* = 0.031), *Erysipelotrichaceae UCG-004* (*p* = 0.047), and *Phocaeicola* (*p* = 0.030) in the rumen. There was also a tendency towards a higher abundance of *Bacteroidales S24-7 group* (*p* = 0.056), *Mogibacterium* (*p* = 0.063), *Defluviitaleaceae UCG-011* (*p* = 0.056), *Eubacterium nodatum group* (*p* = 0.082), *Coprococcus 1* (*p* = 0.056), and *Ruminococcus gauvreauii group* (*p* = 0.090) in the rumen of steers supplemented with energy supplements compared to those that received mineral supplements.

Steers that were supplemented with the PHA showed a lower ruminal abundance of *Ruminiclostridium 6* compared to those that were not supplemented with the PHA (*p* = 0.011). There was a trend for a lower ruminal abundance of *Ruminococcaceae UCG-010* (*p* = 0.089), *Lachnospiraceae uncultured* (*p* = 0.056), and *Marinilabiaceae uncultured* (*p* = 0.075) in steers supplemented with the PHA as compared to those fed supplements without the PHA.

## 4. Discussion

It is well known that supplementation can modulate the forage intake in grazing cattle through associative effects, such as substitutive, additive, or combined, which result from metabolic and digestive interactions [44]. In this study, the energy supplementation trend allowed an additive effect on forage intake, which consequently increased the total DM intake, as well as the intake of nutrients. A similar additive effect was observed in young beef bulls grazing *U. brizantha* cv Xaraés with energy–protein supplementation at a rate similar to that used in this study [45]. Due to the greater input of nitrogen from the energy supplements, there was also a greater excretion of urinary N and a trend toward the greater excretion of N in the feces. However, these steers also had a higher proportion of N retained/N ingested. The efficiency of nitrogen-use in ruminants shows a high variation, ranging from 15% to 40% [46]. Hoffmann et al. [47] reported an efficiency of nitrogen-use of 42.94% in young Nellore bulls grazing intensively managed tropical pastures and supplemented with 0.3% BW, possibly due to greater synchrony between the degradation of carbohydrates and nitrogenous compounds in the rumen.

Steers that were fed energy supplements had a higher total VFA production and a higher proportion of propionate in the rumen, resulting in a lower A:P ratio 3 h after supplementation. This may be related to the increasing Proteobacteria and Spirochaetae phyla, as well as certain bacteria belonging to Firmicutes, such as *Papillibacter*, *Erysipelotrichaceae UCG-004*, *Mogibacterium*, *Defluviitaleaceae UCG-011*, *Eubacterium nodatum group*, *Coprococcus 1*, and *the Ruminococcus gauvreauii group*. This increase suggests an increase in the rumen diet degradation, resulting in higher total VFA production and higher diet digestibility. In addition, some members of Proteobacteria and Spirochaetae phyla are known to be propionate producers [48], while the association of the Firmicutes abundance and the lower A:P ratio in the rumen were previously observed in sheep [49]. Corn gluten meal is an energy supplement ingredient with an energy value equivalent to 92–95% of corn and a higher fermentation rate. It contains high levels of isoleucine (0.73%), leucine (2.15%), and valine (1.19%) [50,51], which could be used as a substrate to propionate production in the rumen. Steers that received energy supplements showed the highest concentration of valerate (1.40%) and isovalerate (2.14%) in their rumen 3 h after supplementation, which may be due to the deamination of these amino acids as a substrate for the formation of branched-chain fatty acids (BCFAs) [52]. A study by Camargo et al. [51] reported that grazing beef cattle supplemented with corn gluten meal had a higher ruminal molar proportion of valerate and isovalerate compared to cattle with ad libitum mineral supplementation. Several ruminal cellulolytic microorganisms, including Ruminococcus, depend on branched-chain fatty acids to grow [48]. This is consistent with the observed increase of the *Ruminococcus gauvreauii group* in the rumen of steers that were given energy supplements.

The inclusion of the PHA in the supplement allowed for a lower ruminal proportion of valerate and tended to increase the ruminal proportion of propionate. The hydrolyzable tannins can modulate protein-degrader microorganisms, decreasing the deamination of soluble protein in the rumen [53] and consequently reducing the formation of BCFA. Similarly, Yang et al. [13] observed that supplying tannic acid at doses of 6.5, 13.0, and 26.0 g/kg DM resulted in an increase in the proportion of propionate and a reduction in the proportion of valerate. These results were also associated by the reduction of enteric methane emissions in cattle. In vitro studies also support the reduction of valerate proportion with increasing doses of tannin extracts while increasing the propionate proportion during rumen fermentation [54]. Furthermore, no effects were detected on ruminal butyrate production, contrary to those observed by Garcia et al. [19] in an in vitro study evaluating the addition of 250 mg/L of carvacrol. However, the concentration of carvacrol used in the present study may not be sufficient to cause this effect.

According to Calsamiglia et al. [18], compounds such as thymol and carvacrol do not selectively inhibit Gram-positive or Gram-negative bacteria. Some specific compounds of essential oils, such as carvacrol, could initiate the loss of cellular content and lysis of Gram-negative bacteria cells due to the presence of the carbonyl group [55]. The inclusion of the PHA in the supplements resulted in a reduction of the ruminal relative abundance of the phylum Verrucomicrobia. This phylum was more abundant in ruminants fed forage diets and its low tolerance to tannins extracts has been previously reported [56,57]. The PHA did not influence the abundance of other phyla, which is contrary to what was observed in grazing goats fed extracts of chestnut and quebracho. In those goats, the extract reduced the abundance of Firmicutes and increased the abundance of Bacteroidetes [58]. The same was observed in grazing cattle that were supplemented with condensed tannins from quebracho and hydrolyzable tannins from *Castanea* spp. The supplementation resulted in a reduction of the relative ruminal abundances of Firmicutes, Fusobacteria, and Fibrobacteres, and an increase in the relative abundances of Armatimonadetes and Actinobacteria [5]. At the family level, the PHA promoted the ruminal abundance of the *Bacteroidales BS11 gut group*, which is a common and abundant group of bacteria found in the rumen. This group of bacteria specializes in fermenting different hemicellulosic monomers to produce acetate and butyrate [59].

In addition, the PHA resulted in a reduction of the genera *Ruminiclostridium 6, Ruminococcaceae UCG-010*, and *Lachnospiraceae uncultured*, which belong to the Firmicutes phylum and Clostridiales order, and are typical of cellulolytic bacteria [60]. This deleterious effect of the PHA was also observed in the rumen of steers supplemented with the EPHA, as they had a lower relative abundance of both *Roseburia, Family XIII UCG-002* genera, as well as the Veillonellaceae family. This reduction may be associated with the lower total digestibility of DM observed with the addition of the PHA in this study and the lower fiber digestibility observed by Teobaldo et al. [22] in a performance trial. Witzig et al. [8] also observed a reduction in the relative abundance of *Ruminococcus albus* by chestnut and valonea tannins. Additionally, the inclusion of hydrolyzable tannins from chestnut tannin extracts reduced the abundance of *Ruminococcus flavefaciens* [6]. In this study, the negative effect of the PHA on the archaea population was observed just on an *uncultured archaeon OTU* belonging to the Thermoplasmatales order. The abundance of *Methanobrevibacter* was reduced when 1.5 g of hydrolyzable tannins were evaluated in vitro [7], indicating that a tannin extract can also affect the archaea population in the rumen. The lower ruminal abundance of the Archaea genus *Methanosphaera* in steers supplemented with the MW may be associated with the fact that *Methanosphaera* uses methyl groups derived from methoxyl substitutes of plant material as the main substrate. Thus, the production of CH_4_ derived from the methyl group is limited by the availability of methyl donors, and the relative abundance of *Methanosphaera* was negatively related to CH_4_ production in sheep [61].

It was expected that the PHA supplements would improve nitrogen utilization in grazing steers, since hydrolyzable tannins can decrease the degradation of soluble protein in the rumen and could potentially improve nitrogen utilization in ruminants [53]. However, in this study, the PHA inclusion did not affect the ruminal concentration of NH_3_-N. Nonetheless, a higher NH_3_-N concentration was observed in the rumen of steers supplemented with energy supplements compared to mineral supplements. This difference in NH_3_-N is attributed to the composition of the energy supplements, as ingredients such as corn gluten meal allowed for a higher CP intake and, consequently, higher protein degradation in the rumen. In this sense, the relationship between the CP content of the diet and the amount of digestible organic matter (DOM) could influence the efficiency of the transfer of ingested CP to the intestine. According to Poppi and McLennan [62], efficient transfer occurs when these values are lower than 160 g CP/kg DOM, and values higher than 210 g CP/kg DOM may result in losses and/or incomplete transfer. In the present study, *U. brizantha* cv. Marandu was intensively managed, allowing CP values above 15% and values higher than 210 g CP/kg MOD (around 254.8 g CP/kg MOD). In addition, serum urea N levels may indicate both the protein status in the animal and the maximum microbial efficiency. According to Vendramini et al. [63], serum urea N levels between 15 and 19 mg/dL are considered threshold levels to prevent protein loss. Thus, the highest ruminal NH_3_-N concentration corroborates the premise of a higher concentration of N-urea in the serum [64]. However, the levels were close to the threshold even in steers that received energy supplements (around 15.98 mg/dL).

The ability to form complexes with proteins, which could lead to a lower ruminal NH_3_-N concentration and a higher flux of rumen undegraded protein to the small intestine, is generally greater for animals supplemented with hydrolyzable tannins with a higher molecular weight [65]. However, neither the Nmic nor the efficiency of the microbial protein synthesis were influenced by the PHA inclusion in the supplements, and the average value of the microbial efficiency observed (15.21 g Nmic/kg OMFR) was below the 30 g Nmic/kg OMFR value recommended by the ARC [35]. The efficiency of microbial protein synthesis details the energy directed toward nitrogen assimilation by rumen microorganisms, and lower values are representative of cattle-fed tropical conditions due to the nutritional unbalance of pastures [23].

## 5. Conclusions

Energy supplementation is an effective strategy for improving the intake and digestibility of forage and nutrients, as well as nitrogen retention in beef steers grazing tropical forage during the rainy season. Energy supplementation modulates the Firmicutes, Protobacteria, and Spirochaetae phyla, resulting in an increase in the production of total volatile fatty acids and the proportion of propionate, valerate, and isovalerate in the rumen.

The inclusion of a blend of phytogenic compounds containing hydrolyzable tannins, carvacrol, and cinnamaldehyde oil in the supplementation of grazing beef steers at a dose of 1.5 g/kg of ingested dry matter did not improve the nitrogen retention or the efficiency of ruminal microbial nitrogen synthesis. Moreover, it reduced the ruminal proportion of valerate and had a negative impact on both the total dry-matter digestibility and the abundance of several ruminal bacterial groups belonging to the Firmicutes and Verrucomicrobia phyla. Further research needs to be conducted to investigate the effects of different doses and types of blends of phytogenic compounds on rumen fermentation, microbiota populations, and nutrient-use efficiency in cattle grazing tropical forages.

## Figures and Tables

**Figure 1 microorganisms-11-00810-f001:**
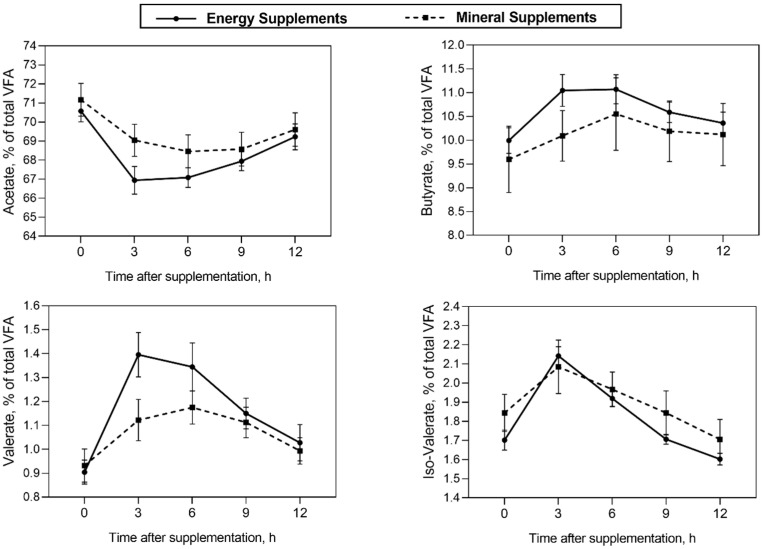
Interaction effect of type of supplement × time after supplementation on the ruminal proportion of acetate (*p* < 0.001), butyrate (*p* < 0.001), valerate (*p* < 0.001), and isovalerate (*p* = 0.008) in Nellore cattle grazing *Urochloa brizantha* cv. Marandu supplemented with energy supplements or mineral supplements.

**Figure 2 microorganisms-11-00810-f002:**
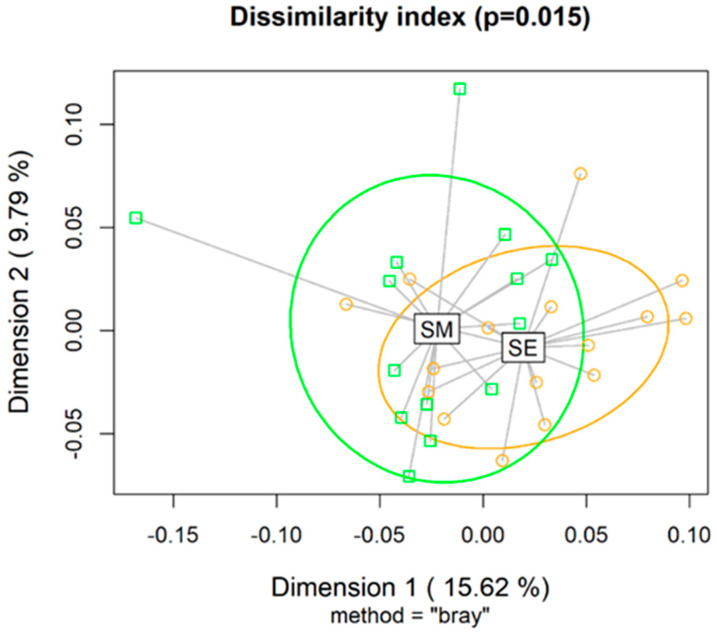
Beta-diversity Bray–Curtis index computed for nonparametric statistical tests between rumen microbial communities of Nellore cattle grazing *Urochloa brizantha* cv. Marandu and supplemented with energy supplements (SEs) or mineral supplements (SMs). There were no observed effects of the addition of a phytogenic compound blend containing 10% of carvacrol and cinnamaldehyde oil, and 90% hydrolyzable tannins extracted from berries and grapes at a dose of 1.5 g/kg of ingested dry matter.

**Figure 3 microorganisms-11-00810-f003:**
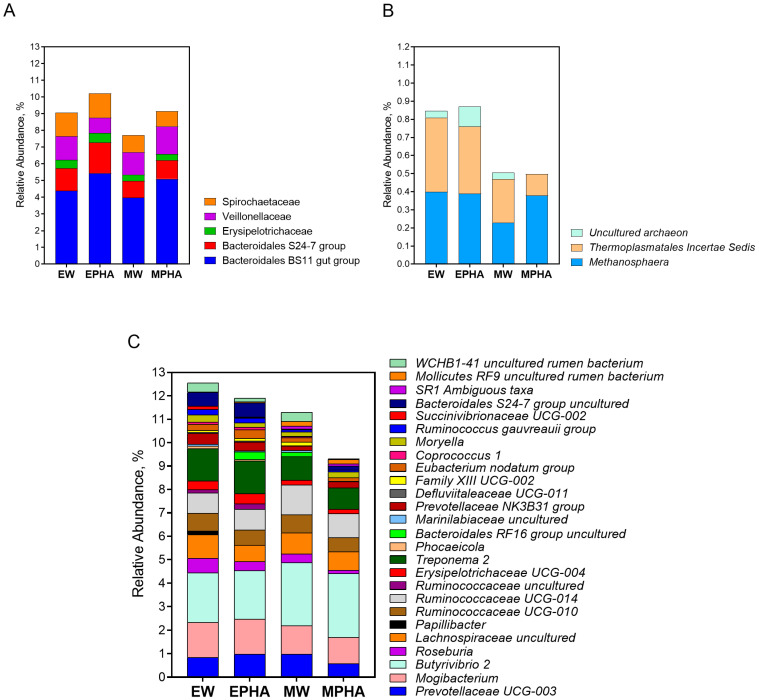
Relative abundance of operational taxonomic units (OTUs) classified at family (**A**) and genera level of ruminal Archaea (**B**) and bacteria (**C**) domain in Nellore cattle grazing *Urochloa brizantha* cv. Marandu during the rainy season influencing by the supplementation with energy supplement without phytogenic compounds addition (EW), energy supplement with the phytogenic compound addition (EPHA), mineral supplement without the phytogenic compound addition (MW), or mineral supplement with the phytogenic compound addition (MPHA). Only the taxa with significative (*p* < 0.05) or tendencies (*p* < 0.10) values are shown.

**Table 1 microorganisms-11-00810-t001:** Chemical composition of forage (*Urochloa brizantha* cv. Marandu) and energy supplements.

Chemical Composition, % DM	Supplements ^1^				Forage ^2^
	EW	EPHA	MW	MPHA	
Mineral matter, %	34.02	33.12	-	-	8.91 ± 0.06
Organic matter, %	65.98	66.88	-	-	91.08 ± 0.07
apNDF ^3^, %	30.32	30.31	-	-	54.85 ± 0.38
iNDF ^4^, %	15.75	15.61	-	-	17.75 ± 0.26
Ether extract, %	1.81	2.11	-	-	2.47 ± 0.05
Gross energy, MJ/kg DM	12.58	13.33	-	-	17.95 ± 0.04
Crude protein (CP), %	14.87	15.02	-	-	15.28 ± 0.22
CNCPS Fraction ^5^, as % of CP					
A	29.01	36.98	-	-	26.64
B1	6.33	6.48	-	-	7.40
B2	50.92	45.01	-	-	45.70
B3	10.89	8.76	-	-	14.20
C	2.85	2.77	-	-	6.06

^1^ EW = energy supplement containing corn gluten meal and minerals without the phytogenic compound addition; EPHA = energy supplement containing corn gluten meal and minerals with the phytogenic compound addition (containing 10% of carvacrol and cinnamaldehyde oil, and 90% hydrolyzable tannins extracted from berries and grapes) at a dose of 1.5 g/kg of ingested DM. MW = mineral supplement warranty levels: Ca, 123.0 g/kg; P, 90.0 g/kg; Cu, 1040.0 mg/kg; Mn, 500 mg/kg; Zn, 2000.0 mg/kg; Co, 15.0 mg/kg; I, 67.0 mg/kg; Se, 14.0 mg/kg. MPHA = guarantee levels similar to DM which include phytogenic compounds at a dose of 1.5 g/kg of ingested DM. ^2^ Chemical composition of samples obtained by simulated grazing technique. ^3^ apNDF: neutral detergent fiber corrected for ash and protein.^4^ iNDF: indigestible neutral detergent fiber. ^5^ The Cornell Net Carbohydrate and Protein System fraction in which A = nonprotein nitrogen, B1 = rapidly degradable protein in the rumen, B2 = moderately degradable protein in the rumen, B3 = slowly degradable protein in the rumen, and C = nondegradable protein in the rumen and unavailable to the animal.

**Table 2 microorganisms-11-00810-t002:** Effect of phytogenic compounds in mineral or energy supplementation on intake and apparent total digestibility of Nellore cattle grazing *Urochloa brizantha* cv. Marandu during the rainy season.

	Supplements ^1^				SEM	*p*-Value ^2^		
	EW	EPHA	MW	MPHA		ST	PHA	ST × PHA
Intake (% BW)								
Total DM	2.52	2.42	1.87	1.97	0.08	0.001	0.977	0.623
Forage DM	2.25	2.15	1.87	1.97	0.07	0.059	0.983	0.474
apNDF	1.30	1.26	1.16	1.10	0.10	0.121	0.595	0.949
Intake (kg/d)								
Total DM	11.60	10.92	8.75	9.15	0.38	0.001	0.810	0.386
Forage DM	10.36	9.68	8.65	9.04	0.32	0.070	0.816	0.392
Supplement DM	1.24	1.23	0.10	0.11	0.02	<0.001	0.747	0.702
OM	10.27	9.63	7.88	8.24	0.33	0.003	0.803	0.417
CP	1.79	1.66	1.39	1.39	0.06	0.006	0.551	0.559
apNDF	6.02	5.66	4.76	5.04	0.19	0.017	0.911	0.381
GE, MJ/d	201.5	189.0	156.4	162.3	6.35	0.004	0.761	0.411
Digestible energy, MJ/d	138.6	125.6	102.8	98.51	5.45	0.002	0.340	0.627
Metabolizable energy, MJ/d	117.6	107.3	85.44	79.43	8.76	0.001	0.321	0.791
g CP/kg DOM	247.5	249.5	259.4	262.8	13.87	0.111	0.721	0.927
Digestibility (%)								
DM	67.0	65.1	62.5	60.6	0.70	<0.001	0.073	0.177
OM	71.0	69.5	64.0	64.1	0.90	<0.001	0.555	0.364
CP	70.0	67.3	58.9	59.7	1.31	<0.001	0.515	0.235
apNDF	72.5	71.9	66.2	67.2	0.91	<0.001	0.877	0.560
GE	68.5	66.3	60.0	60.6	1.02	<0.001	0.536	0.219

^1^ EW = energy supplement containing corn gluten meal and minerals without the phytogenic compound addition; EPHA = energy supplement containing corn gluten meal and minerals with the phytogenic compound addition (containing 10% of carvacrol and cinnamaldehyde oil, and 90% hydrolyzable tannins extracted from berries and grapes) at a dose of 1.5 g/kg of ingested DM. MW = mineral supplement warranty levels: Ca, 123.0 g/kg; P, 90.0 g/kg; Cu, 1040.0 mg/kg; Mn, 500 mg/kg; Zn, 2000.0 mg/kg; Co, 15.0 mg/kg; I, 67.0 mg/kg; Se, 14.0 mg/kg. MPHA = guarantee levels similar to DM include phytogenic compounds at a dose of 1.5 g/kg of ingested DM. ^2^ ST = effect of the type of supplement as mineral or energy supplement. PHA = effect of the addition of the phytogenic compound blend. SEM = standard error of the mean. DM = dry matter, OM = organic matter, CP = crude protein, apNDF = neutral detergent fiber corrected for ash and protein, GE = gross energy, DOM = digestible organic matter.

**Table 3 microorganisms-11-00810-t003:** Effect of phytogenic compounds in mineral or energy supplementation on nitrogen balance and efficiency of ruminal microbial nitrogen synthesis in Nellore cattle grazing *Urochloa brizantha* cv. Marandu during the rainy season.

	Supplements ^1^				SEM	*p*-Value ^2^		
	EW	EPHA	MW	MPHA		ST	PHA	ST × PHA
N balance								
N intake, g/d	286.0	266.1	222.6	222.4	9.69	0.006	0.551	0.559
Fecal N excreted, g/d	89.77	91.59	79.37	84.30	2.94	0.081	0.407	0.906
Urinary N excreted, g/d	116.3 ^a^	96.6 ^ab^	89.7 ^b^	96.1 ^b^	3.93	0.006	0.271	0.033
N retained, g/d	94.51	99.73	54.46	45.82	7.59	<0.001	0.611	0.992
N retained, % of N intake	31.10	33.99	21.97	18.96	2.04	0.050	0.987	0.409
Ruminal microbial N synthesis								
Nmic, g N/d	60.15	54.43	55.92	56.63	6.48	0.848	0.636	0.544
Pmic, g protein/d	375.95	340.18	349.51	353.95	40.47	0.848	0.636	0.544
ENmic, g Nmic/kg OMFR	13.10	13.36	17.39	17.00	2.29	0.055	0.974	0.868
EPmic, g Pmic/kg DOM	53.21	54.28	70.64	69.06	9.29	0.055	0.974	0.868
Plasma urea N ^3^, mg/dL	15.95	16.01	14.52	14.61	0.22	<0.001	0.786	0.939

^1^ EW = energy supplement containing corn gluten meal and minerals without the phytogenic compound addition; EPHA = energy supplement containing corn gluten meal and minerals with the phytogenic compound addition (containing 10% of carvacrol and cinnamaldehyde oil, and 90% hydrolyzable tannins extracted from berries and grapes) at a dose of 1.5 g/kg of ingested DM. MW = mineral supplement warranty levels: Ca, 123.0 g/kg; P, 90.0 g/kg; Cu, 1040.0 mg/kg; Mn, 500 mg/kg; Zn, 2000.0 mg/kg; Co, 15.0 mg/kg; I, 67.0 mg/kg; Se, 14.0 mg/kg. MPHA = guarantee levels similar to DM include phytogenic compounds at a dose of 1.5 g/kg of ingested DM. ^2^ ST = effects of the type of supplement as mineral or energy supplement. ^3^ Time affected the plasma urea concentration (0 h = 11.1 vs. 4 h = 16.43 mg/dL; *p* < 0.001). SEM = standard error of the mean. PHA = effect of the addition of the phytogenic compound blend. Nmic = ruminal microbial nitrogen synthesis, Pmic = ruminal microbial protein synthesis, OMFR = digestible organic matter apparently fermented in the rumen, DOM = digestible organic matter. ^a^,^b^ Values on the same row with unlike superscript letters were significantly different (*p* < 0.05), as obtained with Tukey’s test.

**Table 4 microorganisms-11-00810-t004:** Effect of phytogenic compounds in mineral or energy supplementation on ruminal fermentation parameters in Nellore cattle grazing *Urochloa brizantha* cv. Marandu during the rainy season.

	Supplements ^1^				SEM	*p*-Value ^2^				
	EW	EPHA	MW	MPHA		ST	PHA	Time	ST × PHA	ST × Time
pH	6.25	6.14	6.35	6.34	0.02	0.115	0.691	<0.001	0.679	0.726
NH3-N, mg/dL	17.56	15.46	14.24	14.23	0.46	0.003	0.228	<0.001	0.233	0.149
Total VFA, mmol/L	117.0	112.9	115.8	118.7	1.16	0.723	0.740	<0.001	0.394	0.671
Individual VFA, % of total VFA										
Acetate	68.30	66.74	69.70	68.99	0.15	0.003	0.335	<0.001	0.190	<0.001
Propionate	17.05	16.74	16.34	16.87	0.08	0.010	0.074	<0.001	0.235	0.106
Butyrate	10.66	10.31	10.06	10.16	0.06	0.198	0.792	<0.001	0.360	<0.001
Isobutyrate	0.95	0.93	0.93	1.13	0.01	0.406	0.635	<0.001	0.978	0.458
Valerate	1.22	1.08	1.08	1.05	0.02	0.001	0.015	<0.001	0.149	<0.001
Isovalerate	1.83	1.76	1.89	1.89	0.02	0.058	0.672	<0.001	0.749	0.008
A:P ratio	4.02	3.91	4.29	4.10	0.03	0.006	0.085	<0.001	0.141	0.009

^1^ EW = energy supplement containing corn gluten meal and minerals without the phytogenic compound addition; EPHA = energy supplement containing corn gluten meal and minerals with the phytogenic compound addition (containing 10% of carvacrol and cinnamaldehyde oil, and 90% hydrolyzable tannins extracted from berries and grapes) at a dose of 1.5 g/kg of ingested DM. MW = mineral supplement warranty levels: Ca, 123.0 g/kg; P, 90.0 g/kg; Cu, 1040.0 mg/kg; Mn, 500 mg/kg; Zn, 2000.0 mg/kg; Co, 15.0 mg/kg; I, 67.0 mg/kg; Se, 14.0 mg/kg. MPHA = guarantee levels similar to DM include phytogenic compounds at a dose of 1.5 g/kg of ingested DM. ^2^ ST = effects of the type of supplement as mineral or energy supplement. PHA = effect of the addition of the phytogenic compound blend; no effects or tendencies of the PHA × time and ST × PHA × Time interactions were observed (*p* > 0.10). SEM = standard error of the mean. Nmic = ruminal microbial nitrogen synthesis, Pmic = ruminal microbial protein synthesis, OMFR = digestible organic matter apparently fermented in the rumen, DOM = digestible organic matter.

**Table 5 microorganisms-11-00810-t005:** Effect of phytogenic compounds in mineral or energy supplementation on the abundance of operational taxonomic units (OTUs) classified at phylum level of ruminal bacteria and Archaea domain (expressed as median ± interquartile range) of the in Nellore cattle grazing *Urochloa brizantha* cv. Marandu during the rainy season.

	Supplements ^1^				*p*-Value ^2^		
Domain; Phylum (%)	EW	EPHA	MW	MPHA	ST	PHA	ST × PHA
Archaea; Euryarchaeota	4.84 ± 1.55	4.37 ± 1.32	3.07 ± 3.20	3.80 ± 1.52	0.343	0.782	0.626
Bacteria; Firmicutes	48.88 ± 5.80	46.24 ± 5.12	51.22 ± 2.91	47.99 ± 4.84	0.069	0.418	0.254
Bacteria; Bacteroidetes	27.71 ± 3.53	30.93 ± 4.62	25.87 ± 1.58	27.88 ± 2.90	0.089	0.244	0.248
Bacteria; Proteobacteria	1.42 ± 0.63	1.40 ± 0.39	1.02 ± 0.57	1.05 ± 0.64	0.046	0.828	0.259
Bacteria; Spirochaetae	1.42 ± 0.50	1.45 ± 0.51	1.02 ± 0.44	0.91 ± 0.40	0.022	0.953	0.152
Bacteria; Chloroflexi	1.09 ± 0.40	0.99 ± 0.39	0.83 ± 0.90	0.98 ± 0.63	0.812	0.953	0.953
Bacteria; Actinobacteria	0.98 ± 0.43	0.95 ± 0.33	0.91 ± 0.52	0.88 ± 1.15	0.782	0.984	0.978
Bacteria; Tenericutes	0.74 ± 0.24	0.65 ± 0.11	0.94 ± 0.44	1.03 ± 0.38	0.114	0.664	0.360
Bacteria; SR1 (Absconditabacteria)	0.45 ± 0.36	0.65 ± 0.45	0.77 ± 0.21	0.52 ± 0.45	0.332	0.722	0.584
Bacteria; Verrucomicrobia	0.44 ± 0.30	0.32 ± 0.22	0.51 ± 0.23	0.15 ± 0.43	0.635	0.022	0.133
Bacteria; Fibrobacteres	0.20 ± 0.14	0.29 ± 0.08	0.21 ± 0.21	0.15 ± 0.18	0.275	0.551	0.404

^1^ EW = energy supplement containing corn gluten meal and minerals without the phytogenic compound addition; EPHA = energy supplement containing corn gluten meal and minerals with the phytogenic compound addition (containing 10% of carvacrol and cinnamaldehyde oil, and 90% hydrolyzable tannins extracted from berries and grapes) at a dose of 1.5 g/kg of ingested DM. MW = mineral supplement warranty levels: Ca, 123.0 g/kg; P, 90.0 g/kg; Cu, 1040.0 mg/kg; Mn, 500 mg/kg; Zn, 2000.0 mg/kg; Co, 15.0 mg/kg; I, 67.0 mg/kg; Se, 14.0 mg/kg. MPHA = guarantee levels similar to DM include phytogenic compounds at a dose of 1.5 g/kg of ingested DM. ^2^ ST = effect of the type of supplement as mineral or energy supplement by a Friedman test. PHA = effect of the addition of phytogenic compounds by a Friedman test.

## Data Availability

Data will be made available upon request directly to the authors.

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
