# Peer review of "The Impact of Mineral and Energy Supplementation and Phytogenic Compounds on Rumen Microbial Diversity and Nitrogen Utilization in Grazing Beef Cattle"

_microorganisms, 2023, doi:10.3390/microorganisms11030810_

Round 1

Reviewer 1 Report

The manuscript is interesting and the description of experimental methods and results is detailed and informative. However, there are several issues that need to be addressed before the manuscript can be considered for publication. Firstly, the paper lacks a clear and valuable conclusion. While the experimental data presented is comprehensive, the results are not sufficiently synthesized and analyzed to provide meaningful insights. It is recommended that the author revisit the results section and provide a more in-depth discussion to lead to a significant conclusion. In addition, there are some issues that need to be modified. Specific suggestions are as follows: Introduction: The introduction section of the paper must explicate the interrelationship between the incorporation of energy supplement and trace element supplements and the phytogenic compounds in order to elucidate the underlying purpose of the experiment. The introduction should establish the context of the research, present a clear statement of the research problem or question. Line 90 - 94: Eight steers were assigned to a 4 × 4 Latin square design. I think this paragraph needs to clarify how eight steers were assigned. Line 119: "These supplements" This sentence is prone to ambiguity. Line 120: Corn gluten meal is not the only feed ingredient used in the formula, so there is no need to highlight it. Table 1: "Fraction" Does this refer to the CNCPS fraction? I suggest that this is clearly stated in the table note. Line 140: "on days 16–28 of each experimental period". As described in the previous section, each experimental period lasted 21 days, which seems contradictory. Line 326: "NH3-N" needs subscript. Please scan the manuscript for similar mistakes elsewhere and amend them. Figure 3: Are only the taxa with significant differences present in the Figure? Please clarify it. Line 451 - 487: The analysis of PCA biplot may be inappropriate. In PCA biplot, the correlation coefficient is a measure of the strength and direction of the linear relationship between the variable and the principal component. The correlation coefficient in a PCA biplot is useful for identifying the variables that are most strongly associated with each principal component. However, it simply indicates a statistical association between the variable and the principal component. Principal components do not represent experimental factors, and a strong correlation between a variable and a principal component does not necessarily mean that the variable causes the variation in the data explained by that principal component. It is important to note that the correlation coefficient in PCA biplot does not indicate causation. Line 618: The conclusion may fail to adequately summarize the main findings, and lack clear suggestions for future research. Authors should pay attention to the conclusion section and ensure that it provides a strong summary of this research.

Author Response

The manuscript is interesting and the description of experimental methods and results is detailed and informative. However, there are several issues that need to be addressed before the manuscript can be considered for publication.

Answer: Dear reviewer, we greatly appreciate the reviewer’s efforts for presenting the insightful comments and helpful suggestions. We have made our best efforts to respond to all concerns raised by the reviewer. The original paper has been revised in the light of the comments and suggestions. We hope our revisions have improved the paper to a level of the reviewer’s satisfaction. Point-by-point responses to the comments and suggestions are provided in as follows:

Firstly, the paper lacks a clear and valuable conclusion.

Answer: We appreciate your comment and edit the conclusion.

While the experimental data presented is comprehensive, the results are not sufficiently synthesized and analyzed to provide meaningful insights. It is recommended that the author revisit the results section and provide a more in-depth discussion to lead to a significant conclusion.

Answer: We agree and edited our results and discussion to improved it.

In addition, there are some issues that need to be modified. Specific suggestions are as follows:

Introduction: The introduction section of the paper must explicate the interrelationship between the incorporation of energy supplement and trace element supplements and the phytogenic compounds in order to elucidate the underlying purpose of the experiment. The introduction should establish the context of the research, present a clear statement of the research problem or question.

Answer: Thank you for your comment, we edited the introduction section following your recommendations.

Line 90 - 94: Eight steers were assigned to a 4 × 4 Latin square design. I think this paragraph needs to clarify how eight steers were assigned.

Answer: Dear reviewer, we edited the sentence for more clarity. We used eight steers which were randomly assigned to a double 4 x 4 latin square design. A double 4 x 4 Latin square design means that there are two Latin squares used in the experiment, each with four rows (periods) and four columns (treatments).

Line 119: "These supplements" This sentence is prone to ambiguity.

Answer: We edited it.

Line 120: Corn gluten meal is not the only feed ingredient used in the formula, so there is no need to highlight it.

Answer: we agree and edited it.

Table 1: "Fraction" Does this refer to the CNCPS fraction? I suggest that this is clearly stated in the table note.

Answer: Yes, it is, we agree with the suggestion and stated it in the table.

Line 140: "on days 16–28 of each experimental period". As described in the previous section, each experimental period lasted 21 days, which seems contradictory.

Answer: You are totally right. We apologize by the typing error. Our experimental periods lasted 28 days. We correct it on M&M. Thanks for catching.

Line 326: "NH3-N" needs subscript. Please scan the manuscript for similar mistakes elsewhere and amend them.

Answer: thank you for catching it. We checked our manuscript and corrected several mistakes.

Figure 3: Are only the taxa with significant differences present in the Figure? Please clarify it.

Answer: Yes, it is. We edit the caption figure to clarify it.

Line 451 - 487: The analysis of PCA biplot may be inappropriate. In PCA biplot, the correlation coefficient is a measure of the strength and direction of the linear relationship between the variable and the principal component. The correlation coefficient in a PCA biplot is useful for identifying the variables that are most strongly associated with each principal component. However, it simply indicates a statistical association between the variable and the principal component. Principal components do not represent experimental factors, and a strong correlation between a variable and a principal component does not necessarily mean that the variable causes the variation in the data explained by that principal component. It is important to note that the correlation coefficient in PCA biplot does not indicate causation.

Answer: Authors agree and remove this analysis.

Line 618: The conclusion may fail to adequately summarize the main findings, and lack clear suggestions for future research. Authors should pay attention to the conclusion section and ensure that it provides a strong summary of this research.

Answer: We agree and edited the conclusion. Thanks for the suggestion.

Reviewer 2 Report

In this paper, the authors focus on the use of feed additives as a method for altering rumen fermentation through altering the rumen microbiota.

General comments:

After reading the abstract where the authors state that there is no effect of phytogenic compounds on the nitrogen utilization, I find the title of the manuscript misleading and requires change.

 In the introduction, the clear gap of knowledge is not defined and the role of diet in the potential interactions is also ignored. As well it is unclear to the reader why energy vs. mineral is being tested. 

In the materials and methods, the authors state that mineral was fed ad libitum, which would imply that the phytogenic products were not given equally and no measurement of intake was possible, which would be a large issue for the determination of impact for this research. As well, the method of bacterial pellet extraction is not described which would have a huge impact on the data analysis. 

In the results, it seems as though all significant results are related to the presence of corn in the supplement. This is a large confounding factor in the data, however, authors do not address this issue adequately in the discussion. 

Picrust analysis is weak in 16S, therefore, it is necessary to provide the value for NSTI to show the coverage database. 

Overall the discussion extensively discusses types of microbial changes without clearly making associations between bacteria and their impact on rumen metabolism in the experiment. 

Line 100: what is the put and take method

Line 221: primers must be listed. 

Table 2 - why is rainy season necessary - what does this mean for the data interpretation

Table 3 - what are the letters in the table, why are there only letters for Urinary N - is it associated with the interaction? What about where there was no interaction.

Figure 3 - not legible. 

Figure 4 - why is it labelled protein source?

Author Response

In this paper, the authors focus on the use of feed additives as a method for altering rumen fermentation through altering the rumen microbiota.

Answer: Dear reviewer, we greatly appreciate the reviewer’s efforts for presenting the insightful comments and helpful suggestions. We have made our best efforts to respond to all concerns raised by the reviewer. The original paper has been revised in the light of the comments and suggestions. We hope our revisions have improved the paper to a level of the reviewer’s satisfaction. Point-by-point responses to the comments and suggestions are provided in as follows:

General comments:

After reading the abstract where the authors state that there is no effect of phytogenic compounds on the nitrogen utilization, I find the title of the manuscript misleading and requires change.

Answer: Authors agree and modified the tittle.

In the introduction, the clear gap of knowledge is not defined and the role of diet in the potential interactions is also ignored. As well it is unclear to the reader why energy vs. mineral is being tested.

Answer: We appreciated your comment and edit our introduction to clarified it.

In the materials and methods, the authors state that mineral was fed ad libitum, which would imply that the phytogenic products were not given equally and no measurement of intake was possible, which would be a large issue for the determination of impact for this research.

Answer:  Dear reviewer, we understand your concern. In general, in pastures of high nutritional value, as in this experiment, the animals consume up to 100 g/day of mineral salt. Remembering that NaCl is an intake restrictor, thus is not possible that bovine eats 0.3% of salt. Mineral supplements are salts (no energy, no cp or N input) for this reason are provided ad libitum. The mineral mixture is a common practice used in mineral nutrition, thus ensuring the consumption of macro and micronutrients. Brazil is one of the largest beef cattle producers in the world, and the pasture-based system represents the bulk of its beef cattle production system.  The beef cattle production in Brazil is centered on the use of tropical grasses. The mineral supplementation is the business-as-usual practice in the beef cattle production in the studied regions. Therefore, the control treatment is pasture + mineralization. Mineral supplementations refer to the use of microelements (Calcium and Phosphorus) and microelements (Copper, Manganese, Zinc, Cobalt, Iodine and Selenium). We simulate two real situations in the productive systems (mineral supplementation or energy supplementation).

As well, the method of bacterial pellet extraction is not described which would have a huge impact on the data analysis.

Answer: Thank you for your comment, we added a brief description of pellet formation.

In the results, it seems as though all significant results are related to the presence of corn in the supplement. This is a large confounding factor in the data, however, authors do not address this issue adequately in the discussion.

Answer: dear reviewer, our experimental design was a double 4 × 4 Latin square design in a factorial arrangement (A × B), where we considered the factor A to evaluate the type of supplement effect, as mineral or energy supplement, and the factor B to evaluate the phytogenic compounds blend (PHA) addition (yes or no). In addition, we also included the factors (A×B) interactions evaluation. For this reason we included in our tables results three p-values, one per each factor (as ST and PHA) and one about the interaction factors (as ST X PHA). You are right, mostly of our results showed that the type of supplement (mineral vs energy) affected our variables answer. And that the PHA had minimal effects. A factorial arrangement (A × B) is a statistical design used to study the effects of two factors, A and B, on a response variable. In this design, each level of factor A is combined with each level of factor B to create a set of unique treatment combinations. Factorial arrangement is a powerful statistical tool for investigating the effects of multiple factors on a response variable, and it allows for a more efficient use of resources compared to studying each factor individually.

Picrust analysis is weak in 16S, therefore, it is necessary to provide the value for NSTI to show the coverage database.

Answer: Considering your comment and concerns of the other reviewer about PCA analysis we choice to remove these data.

Overall the discussion extensively discusses types of microbial changes without clearly making associations between bacteria and their impact on rumen metabolism in the experiment.

Answer: we agree and edited the discussion section.

Line 100: what is the put and take method

Answer: The Put and Take method described by Mott and Lucas is a simple but effective way to estimate the amount of forage consumed by grazing animals. The Put and Take method consists of using additional animals to tester animals that remain in the pasture throughout the experimental period. Weight gain is evaluated in test animals, while put and take methods are used, calculating the number of animals/day kept in the area. Then we add the days of the testers to the animals and we have the total number of animal days in the period. Considering the weight, we have the AU/ha. This method was described by Mott, G.O.; Lucas, H.L. The design conduct and interpretation of grazing trials on cultivated and improved pastures. In Proceedings of the 6th International Grassland Congress, State College, PA, USA, 17–23 August 1952.

Line 221: primers must be listed.

Answer: we listed it.

Table 2 - why is rainy season necessary - what does this mean for the data interpretation

Answer: Thank for you your question. The beef cattle production in Brazil is centered on the use of tropical grasses. In the rainy season these forages reach maximum production (70 to 80% of the total forage produced annually) and nutritional quality compared to the dry season. The rainy season is the period that favours the growth of tropical grass, as there is no restriction of humidity, light, and high temperatures for the growth of forage. In this region of Brazil, the winter period in which we have temperatures below 15 ºC, associated with dry, reduces forage production. We have only 20% of the total production in winter.

Table 3 - what are the letters in the table, why are there only letters for Urinary N - is it associated with the interaction? What about where there was no interaction.

Answer: The superscript letter indicating that the factor ST X PHA interaction was significative and allows to the reader identified which treatment was higher or lower than. For this reason, only are in the Urinary N excreted, g/d (the only one variable affected significatively by the ST X PHA interaction). We added a description into footnote of table for more clarity. When there was no interaction, means that the p-value of individual factors (ST or PHA) need to be considered, and if significative, in the text of results could find the description about the difference.

Figure 3 - not legible.

Answer: We edited the figure to improve it.

Figure 4 - why is it labelled protein source?

Answer: Is was a typing error in the script of PCA biplot.

Round 2

Reviewer 1 Report

The author has provided clear and concise responses that address all the concerns and have improved the quality of the paper.

Reviewer 2 Report

The authors have adequately addressed all issues.